# LTLBench: Towards Benchmarks for Evaluating Temporal Reasoning in Large Language Models

## Abstract

Temporal Reasoning (TR) is a critical ability for LLMs to understand and reason over temporal information and relationships between events. To study the TR ability in LLMs, prior works provide different ways for evaluating various aspects of TR ability. In this work, we propose an alternative perspective for evaluating TR ability by leveraging Linear Temporal Logic (LTL), and develop a pipeline to automatically synthesize challenges for assessing the TR ability of LLMs. Based on this pipeline, we construct a dataset, namely LTLBench, consisting of 2000 TR challenges, and benchmark 12 LLMs across 5 different methods. Furthermore, we conduct additional experiments to investigate the impact of increasing the number of formula operators and events on both LLM performance and the complexity of TR problems. We also perform qualitative analyses of their reasoning processes and the effects of varying the number of events and formula operators, which reveal 3 main issues in their temporal reasoning processes and the unexpected performance changes observed as problem complexity increases. We expect this work to provide valuable insights into the TR ability of LLMs[1].

## 1 Introduction

Temporal Reasoning (TR) is a critical reasoning ability of LLMs, encompassing the understanding, processing, and reasoning over temporal information and relationships between events, which is essential for solving problems across diverse scenarios (Shoham & Goyal, 1988; Chittaro & Montanari, 2000; Vila, 1994). Prior studies have demonstrated that although LLMs show some promise in TR, they still struggle with TR and a substantial performance gap persists between the state-of-the-art LLMs and humans on TR (Chu et al., 2023; Wang & Zhao, 2024; Beniwal et al., 2024). Nevertheless, since TR encompasses various aspects, existing investigations of the TR ability of LLMs remain incomplete. Therefore, in this work, we adopt Linear Temporal Logic to explore TR from a new perspective, focusing on formal logical reasoning over temporal information.

Linear Temporal Logic (LTL) is a formal logic, specifically a modal temporal logic, that is widely used and studied for expressing and reasoning about sequences of events over time (Kröger & Merz, 2008; Goranko & Rumberg, 2024). Although complex LTL is typically discussed and used in the context of formal and program verification, basic and moderately complex LTL tasks are, in fact, ubiquitous in daily tasks. For example, if people are out of milk, they will eventually buy it, which can be formalized in an LTL formula as $G(\text{OutOfMilk} \rightarrow F(\text{BuyMilk}))$. Likewise, if the traffic light is green, it will then turn to yellow, which can be formalized as $G(\text{Green} \rightarrow X(\text{Yellow}))$. From this perspective, LTL provides a natural and formal way to represent and operate on temporal relations between events. Therefore, we leverage LTL as a main component to construct TR problems.

To explore the TR ability of LLMs, we propose a novel approach, a TR challenges generation pipeline, to automatically synthesize TR problems. Each generated TR problem mainly consists of a context that depicts the situation of a TR problem and a hypothesis that requires LLMs to judge its validity against the given context. The core components of the pipeline involve a randomly generated directed graph that serves as

---

[1]Our code is open-sourced and available at `https://anonymous.4open.science/r/LTLBench-Anonymous/`.

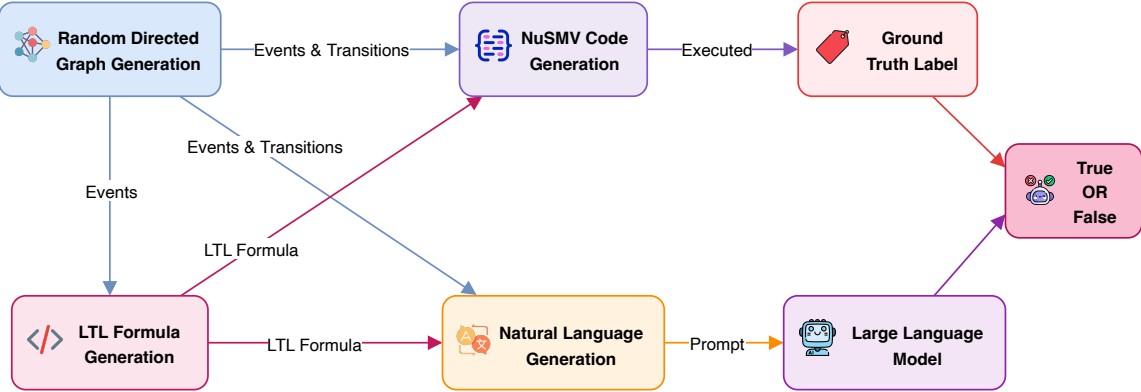

Figure 1: The overview of the TR problem generation pipeline.

the preparation for subsequent problem generations, a randomly generated LTL formula that acts as the hypothesis for the given context, and the NuSMV model checker (Cimatti et al., 2002) which executes the code of events transitions and the LTL formula to provide the ground truth label. As shown in Figure 1, during the generation process for a TR problem, we first generate a random directed graph. Then, we adopt and slightly modify the LTL formulas generation algorithm designed by Zhu (2021) to generate an LTL formula based on the events given in the graph. Subsequently, both the event information and the LTL formula are converted into NuSMV (Cimatti et al., 2002) code and executed to obtain the ground truth label. Finally, the event information and the LTL formula are translated into a TR problem in the form of natural language.

Furthermore, to conduct an intensive and comprehensive evaluation, we generate a total of 2000 TR challenges using our proposed pipeline, referred to as LTLBench, and we evaluate 12 LLMs across 5 different methods. We not only demonstrate their TR ability but also provide several qualitative insights that reveal 3 main issues they fail to reason over these challenges. In addition, we conduct additional experiments to investigate the impact of varying the number of events and operators on both the performance of LLMs and the complexity of TR problems. The key contributions of our study are summarized as follows:

1. We develop a novel TR challenges generation pipeline, which lays on Linear Temporal Logic, to evaluate the TR ability of LLMs from a new perspective, focusing on formal logical reasoning over temporal information;

2. Based on the pipeline, we construct a dataset, LTLBench, consisting of 2000 TR challenges, and evaluate 12 LLMs across 5 different methods, providing both quantitative results and qualitative insights which reveal 3 main issues of their reasoning failures;

3. We further conduct two additional experiments to demonstrate that increasing the number of formula operators and events leads to more challenges for LLMs, and offer qualitative insights to discuss the unexpected performance changes observed as problem complexity increases.

## 2 Related Work

### 2.1 TR in LLMs

Temporal Reasoning in LLMs has recently obtained substantial attention (Xiong et al., 2024; Fatemi et al., 2024; Beniwal et al., 2024; Chu et al., 2023; Hu et al., 2023; Liu et al., 2023; Vashishtha et al., 2020). Beniwal et al. (2024) point out notable deficiencies of LLMs in understanding and reasoning over temporal information and reasoning, while Xiong et al. (2024) propose TG-LLM, a framework aimed at improving LLMs performance on TR tasks. Furthermore, Liu et al. (2025) introduce a Time-R1, which employs

reinforcement learning fine-tuning to enhance temporal reasoning ability of LLMs, enabling smaller models to match or even surpass larger ones. While these works collectively highlight the progress and importance of TR ability in LLMs, the exploration of TR ability in LLMs remains ongoing, requiring further efforts to reveal and understand the boundaries.

## 2.2 TR Benchmarks

To discover and evaluate the TR ability of LLMs, a variety of TR datasets and benchmarks have been proposed, targeting different aspects of TR at varying levels of complexity (Fatemi et al., 2024; Wang & Zhao, 2024; Xiong et al., 2024; Beniwal et al., 2024; Qin et al., 2021; Tan et al., 2023; Virgo et al.). For example, Xiong et al. (2024) construct a TR dataset by leveraging a large temporal knowledge graph, YAGO11k (Dasgupta et al., 2018), and utilizing GPT-3.5 and rule-based Python scripts to generate TR challenges. Additionally, Fatemi et al. (2024) employ random graph generation as a foundation and preparation to form rule-based and different types of temporal facts and questions without introducing LLMs to generate TR tasks, focusing on temporal semantics and arithmetic reasoning and proposing a benchmark called Test of Time. Furthermore, Wang & Zhao (2024) introduce a TR dataset consisting of various temporal aspects such as order, arithmetic, frequency, and duration. While these approaches provide valuable perspectives for evaluating TR ability, they generally lack systematic support for representing and reasoning with temporal operators such as *eventually* and *always*, as well as their complex compositions. By contrast, our work that leverages LTL enables a natural support over those temporal operators and can explore TR ability from another perspective.

## 3 TR Problem Generation Pipeline

The pipeline to generate a single TR problem consists of four stages: (1) Random Directed Graph Generation, (2) LTL Formula Generation, (3) NuSMV Code Generation, and (4) Natural Language Generation. We demonstrate the overview of the process for a TR problem generation in Figure 1.

## 3.1 Random Directed Graph Generation

During this stage, a directed graph is randomly generated with a given number of events $n$ where $n > 1$ to ensure the formation of transitions between events. Specifically, to construct the directed graph, since our pipeline does not impose any particular prior structures or requirements on transitions between events, we adopt a typical and simple random graph generation model, i.e., Erdős–Rényi (ER) random directed graph model $G(n, p)$ with edge probability $p = 0.5$, by which the directed edges of the generated random directed graphs are independently present or absent with equal probability, and also the generated graphs have no preference for sparser or denser transitions structures.

In this graph, each $node_i$ represents an individual $event_i$, and each $edge_j^i$ is a directed edge pointing from $node_i$ to another $node_j$, which forms the relationships and transitions between events, indicating that $event_j$ represented by $node_j$ occurs after the $event_i$ represented by $node_i$. It is important to note that each $node_i$ within the graph can have multiple outgoing edges, signifying that several subsequent events can follow $event_i$, as well as multiple incoming edges, indicating that $event_i$ can be preceded by several other events. To form a clear semantics aligned with the follow-up stages, for an event $event_i$ followed by only one event $event_j$, we say that after $event_i$, $event_j$ must happen, while for an event $event_i$ followed by more than one event $event_1, event_2, \ldots, event_n$, we say that after $event_i$, either $event_1, event_2, \ldots,$ or $event_n$ must happen.

As an example illustrated in Figure 2, given $n = 3$, three events are generated: $event_1$, $event_2$, and $event_3$. The case that $event_1$ points to $event_2$ indicates that $event_2$ must happen after $event_1$. The case that $event_1$ points to $event_3$ and $event_3$ also points to $event_1$ means that $event_1$ must happen after $event_3$ and $event_3$ must happen after $event_1$. In addition, to note, $event_1$ not only points to $event_2$ but also $event_3$, indicating that either $event_2$ or $event_3$ must happen after $event_1$. By contrast, if $event_1$ only points to $event_2$, it indicates that $event_2$ must happen after $event_1$.

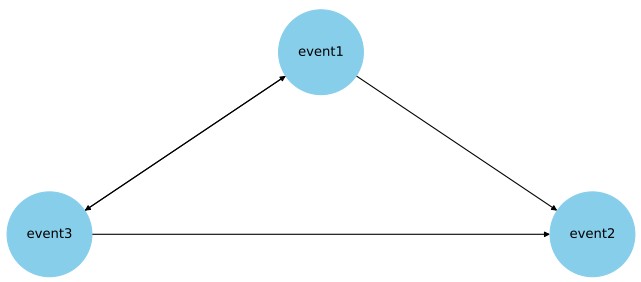

Figure 2: An example of a generated random directed graph.

```
1  (event1 -> (G (F event2)))
```

Listing 1: An example of a generated LTL formula.

The generation of the random directed graph during this stage serves as the foundation and preparation for generating the LTL formula, NuSMV code, and TR problem represented in natural language by providing the information of events and also the transitions between events.

### 3.2 LTL Formula Generation

Based on the events generated in the graph, we employ the algorithm designed by Zhu (2021), as shown in Algorithm 1 in Appendix A.3, to generate an LTL formula with a given number of operators $m$ where $m > 0$. The LTL operators include unary and binary operators. Unary operators, for example, include but are not limited to $X$ which indicates that for a given event $\phi$, $X\phi$ denotes that the event $\phi$ will occur at the next moment, and $F$ for which $F\phi$ means that event $\phi$ will eventually occur at some point in the future. Binary operators include but are not limited to & representing logical AND and | representing logical OR. The given number of operators refers to the number of unary and binary LTL operators contained in an LTL formula.

An example of a generated LTL formula is also provided and shown in Listing 1. The formula means that if $event_1$ happens, it will be globally true that, at some point in the future, $event_2$ will eventually happen.

The generation of the LTL formula aims to be the preparation for generating the LTLSPEC part of the NuSMV code and also to generate the hypothesis part of the TR problem represented in natural language.

### 3.3 NuSMV Code Generation

Given the information of events from the graph and the LTL formula, this stage converts them into NuSMV code, which consists of two parts: (1) context generation and (2) LTLSPEC generation. The context describes the situation of the TR problem, while LTLSPEC represents a hypothesis regarding the context.

For context generation, it includes event definitions, initial event setup, and event transitions setup. Based on the generated graph, events and their transitions are converted into the context part of the NuSMV code, and an initial event is randomly selected. As shown in Listing 2, Lines 2-3 define three events, Line 5 specifies the initial event which is $event_3$, and Lines 6-10 construct the event transitions, in which, for example, Line 7 indicates that $event_2$ and $event_3$ can follow $event_1$, while $event_2$ remains to itself and after $event_2$, no other events can happen.

For LTLSPEC generation, the generated LTL formula is translated into the NuSMV code, as illustrated at Line 11 in Listing 2 which represents the LTL formula shown in Listing 1.

```
1  MODULE main
2  VAR
3      state : {event1, event2, event3};
4  ASSIGN
5      init(state) := event3;
6      next(state) := case
7          state = event1 : {event2, event3};
8          state = event2 : event2;
9          state = event3 : {event1, event2};
10     esac;
11 LTLSPEC ((state=event1) -> (G (F (state=event2))))
```

Listing 2: An example of NuSMV code.

```
1  === Context ===
2
3  Initially, event3 happened. After event1, either event2 or event3 must happen. After event2,
   ↪   no other events can happen. After event3, either event1 or event2 must happen.
4
5  === Hypothesis ===
6
7  C1: Event2 eventually happens.
8  C2: C1 always holds.
9  C3: If event1 happens, then C2 holds.
10
11 C3 is True or False?
```

Listing 3: An example of TR problem in the form of natural language.

The resulting NuSMV code is executed by the NuSMV model cheker during the generation process to obtain the ground truth label for the TR problem.

### 3.4 Natural Language Generation

During this stage, the events information in the graph and the LTL formula are converted into natural language. Similar to the NuSMV code generation, this stage consists of two parts: (1) context generation and (2) hypothesis generation. The context describes the problem situation while the hypothesis specifies what LLMs are required to determine given the context.

For context generation, based on the generated graph and the initial event, event information and transitions are converted into natural language by Algorithm 2 shown in Appendix A.3. As the example shown at Lines 1-4 in Listing 3, the initial event and events transitions are represented in natural language.

For hypothesis generation, the generated LTL formula is transformed into natural language by Algorithm 3 shown in Appendix A.3. As the example shown at Lines 5-10 in Listing 3, the hypothesis is decomposed and represented in natural language. Additionally, Line 11 is used to prompt LLMs to determine the validity of the hypothesis.

The TR problem represented in natural language generated in this stage is the core and final product of the generation process, which will be used to evaluate the TR ability of LLMs.

## 4 Experiment Settings

To evaluate the TR ability of LLMs, based on the pipeline, we construct a dataset, LTLBench, consisting of 2000 problems[2]. Each problem is generated with a fixed number of events $n = 3$ and formula operators $m = 3$. Additionally, to explore the impact of changes in the number of formula operators, we conduct evaluations on newly generated problems with a fixed number of events $n = 2$ while varying the number

---

[2] We detail 5 statistics to provide more characterization of LTLBench in Appendix A.4.

of formula operators $m \in \{1, 2, 3, 4, 5, 7, 9\}$. For each $(n, m_i)$, such as $(2, 1)$ indicating that the number of events is 2 and the number of operators is 1, we generate 300 problems as a dataset for evaluation. Similarly, to examine the effects of the number of events, we fix the number of formula operators $m = 2$ and vary the number of events $n \in \{2, 3, 4, 5, 7, 9\}$. For all generated datasets, their distributions of ground truth labels are meticulously balanced, with half of the problems labeled as *True* and the other half of the problems as *False*. Furthermore, we mainly adopted Accuracy, defined as the proportion of the correctly answered problems, as the primary evaluation metric.

For comprehensive evaluations, we select a total of 12 LLMs. Specifically, we evaluate DeepSeek-V3 (DeepSeek-AI et al., 2025) in both non-thinking mode and thinking mode, which are officially noted as *DeepSeek-Chat* model and *DeepSeek-Reasoner* model. We also include 3 OpenAI models, which are *GPT-3.5-Turbo* (Brown et al., 2020), *GPT-4o-Mini* (OpenAI et al., 2024), and *GPT-5-Mini*. In addition, we select 4 Qwen models (Yang et al., 2025; Qwen et al., 2025), namely *Qwen-Turbo*[3], *Qwen3:32B*, *Qwen3:14B*, and *Qwen2.5:72B-Instruct*. Furthermore, we also include 3 additional models, which are *Gemma3:12B-Instruct* (Team et al., 2025), *Mistral:7B-Instruct* (Jiang et al., 2023), and *Phi4:14B* (Abdin et al., 2024).

Moreover, to examine how different methods affect LLMs performance on TR challenges, we adopt 5 methods, which are *Direct Prompting*, *Zero-Shot CoT* (Kojima et al., 2022), *Few-Shot CoT* (Wei et al., 2022), *Self-Consistency* (Wang et al., 2023), and *Least-to-Most* (Zhou et al., 2022). *Direct Prompting* directly queries an LLM to generate answer in *True* or *False* for a given TR problem. *Zero-Shot CoT* prompts an LLM to reason step by step without examples, whereas *Few-Shot CoT* provides 2 examples with each consisting of a TR problem, thinking process, and final answer. *Self-Consistency* extends *Zero-Shot CoT* by sampling 3 reasoning paths and taking majority voting to select the final answer. In *Least-to-Most*, an LLM is first prompted to answer a set of breakdown smaller questions and is then prompted again with the response to previous questions to reason on the main problem to give the final answer. For *Direct Prompting*, *Zero-Shot CoT*, *Few-Shot CoT*, and *Least-to-Most* methods, we set the *temperature* to 0 and the *max completion tokens* to 2000 for all models. For *Self-Consistency*, since it requires diversity of candidates, we set the *temperature* to 0.7 and the *max completion tokens* to 2000 for all models[4]. The prompt templates for *Direct Prompting*, *Zero-Shot CoT*, and *Few-Shot CoT* are provided in Prompt Template 1, 2, and 3 in Appendix A.1, respectively, with *Self-Consistency* using the same prompt template as *Zero-Shot CoT*. The prompt templates for the two stages of *Least-to-Most* are shown in Prompt Template 4 and 5 in Appendix A.1.

## 5 Results and Analyses

### 5.1 Evaluation with LTLBench

We evaluate the selected 12 LLMs on LTLBench consisting of 2000 generated TR problems with the number of events $n = 3$ and formula operators $m = 3$. The results of all models equipped with different methods are reported in Table 1. We also demonstrate the performance of each model with different methods in a grouped horizontal bar chart as shown in Figure 3 for a better visual comparison of the results.

From the results, among all models and methods, the best one is the *GPT-5-Mini* with the *Few-Shot CoT* method, achieving an accuracy of 93.95%, whereas the worst one is the *GPT-3.5-Turbo* with the *Least-to-Most* method, whose accuracy of 51.05% is only slightly above random guessing. In addition, the average accuracies of each method across models are 65.83% for *Direct Prompting*, 67.18% for *Zero-Shot CoT*, 76.33% for *Few-Shot CoT*, 68.29% for *Self-Consistency*, and 68.21% for *Least-to-Most*, in which *Few-Shot CoT* demonstrates its significant performance improvement while the other three methods slightly outperform the *Direct Prompting*. As demonstrated in both Table 1 and Figure 3, *Few-Shot CoT* consistently and significantly outperforms other methods regardless of models. In addition, *Self-Consistency* tends to fail to further improve the performance of recent high-performing models regardless of the parameter sizes, such as *DeepSeek-Reasoner*, *DeepSeek-Chat*, *GPT-5-Mini*, *GPT-4o-Mini*, *Qwen-Turbo*, *Qwen3:32B*, *Qwen3:14B*, and *Phi4:14B*, while for earlier models such as *Mistral:7B-Instruct*, *GPT-3.5-Turbo*, and *Qwen2.5:72B-Instruct*, it can still significantly enhance their performance. We consider the reason is that after multiple iterations of

---

[3]The exact version of *Qwen-Turbo* is *Qwen-Turbo-2025-04-28*.
[4]For the GPT-5 series models, the *temperature* is not settable, so we do not use *temperature* for *GPT-5-Mini*.

Table 1: The accuracy (%) of LLMs equipped with different methods evaluated on LTLBench, in which DP stands for *Direct Prompting*, ZS CoT for *Zero-Shot CoT*, FS CoT for *Few-Shot CoT*, SC for *Self-Consistency*, and L2M for *Least-to-Most*, and the best accuracy across models for each method is highlighted in bold.

| Model | DP | ZS CoT | FS CoT | SC | L2M |
|---|---|---|---|---|---|
| DeepSeek-Reasoner | 71.55 | 69.95 | 81.10 | 73.70 | 72.75 |
| Deepseek-Chat | **80.10** | **78.65** | 90.40 | 79.40 | **81.05** |
| GPT-5-Mini | 79.70 | 78.45 | **93.95** | **80.53** | 77.75 |
| GPT-4o-Mini | 63.50 | 62.60 | 71.55 | 63.25 | 63.70 |
| GPT-3.5-Turbo | 53.25 | 56.55 | 60.40 | 57.10 | 51.05 |
| Qwen-Turbo | 66.50 | 67.60 | 78.70 | 68.90 | 69.60 |
| Qwen2.5:72B-Instruct | 59.05 | 68.10 | 76.05 | 67.75 | 69.90 |
| Qwen3:32B | 66.45 | 66.65 | 76.50 | 68.10 | 70.45 |
| Qwen3:14B | 67.45 | 67.45 | 75.15 | 67.80 | 71.25 |
| Phi4:14B | 66.65 | 65.60 | 73.50 | 65.90 | 66.15 |
| Gemma3:12B-Instruct | 60.15 | 67.20 | 78.40 | 68.20 | 67.20 |
| Mistral:7B-Instruct | 55.60 | 57.35 | 60.25 | 59.00 | 57.70 |

LLMs, recent LLMs approach a problem in a more self-consistent way, so that the *Self-Consistency* method may fail to further improve the performance on these models. In addition, across most models, *Zero-Shot CoT* can only marginally increase the performance, which could be attributed to that most of the evaluated models have already incorporated reasoning or thinking instructions during their pretraining or instruction-tuning process, such as DeepSeek series models, OpenAI's GPT-5 series models, and also Qwen3 series models, so that even in the case of using *Direct Prompting*, they automatically invoke the chain-of-thought to reason over the problems, as our observation of their responses.

In addition, notably, we observe that *DeepSeek-Chat* outperforms *DeepSeek-Reasoner*. Typically, we believe models equipped with enhanced and polished reasoning ability should outperform ones without it. From our perspective, this unexpected result may arise because our LTL-based TR challenges require a more complex formal logic reasoning along with temporal reasoning, in which the reason steps may be different from other challenges without formal logic injected in such as Test of Time (Fatemi et al., 2024) and may cause untypical reasoning steps. The reasoning steps that *DeepSeek-Reasoner* are trained on during pretraining and perform during inference may not align with the reasoning steps demanded by LTL-based TR problems. Thereby, although it is explicitly asked to think in a specific way to tackle LTL-based TR problems, the pretrained distribution of output tokens of reasoning steps is not suitable for this kind of problem and could even further disturb the reasoning steps to some extent, leading to even worse than equipping non-thinking or non-reasoning models with explicit task-specific CoT methods.

Moreover, to get more insights into the reasoning process of LLMs on the TR challenges, we further conduct a qualitative analysis. For each model paired with each method, we randomly select 10 problem and answer pairs, thus resulting in a total of 600 pairs. We then filter out all pairs with correct answers and finally retain 145 pairs with incorrect answers, to focus on analyzing the potential factors that lead LLMs to fail in answering problems correctly. Among these 145 pairs, we identify three major findings. The first is ***Temporal Semantics and Reasoning Misalignment***, which indicates that although LLMs demonstrate an understanding of temporal semantics during reasoning steps, they may still apply these semantics incorrectly when performing actual reasoning. A representative example is observed in the results of *GPT-4o-Mini* with the *Few-Shot CoT* method, in which although it understands that after *event 1*, the *event 3* must happen, which is then followed by *event 2*, when it reasons about whether *event 2* happens at next state after *event 1*, it does not precisely capture that the exact next state is *event 3* but it reasons in a misaligned way such that since that *event 1* is followed by *event 3* and *event 3* is followed by *event 2*, *event 2* happens

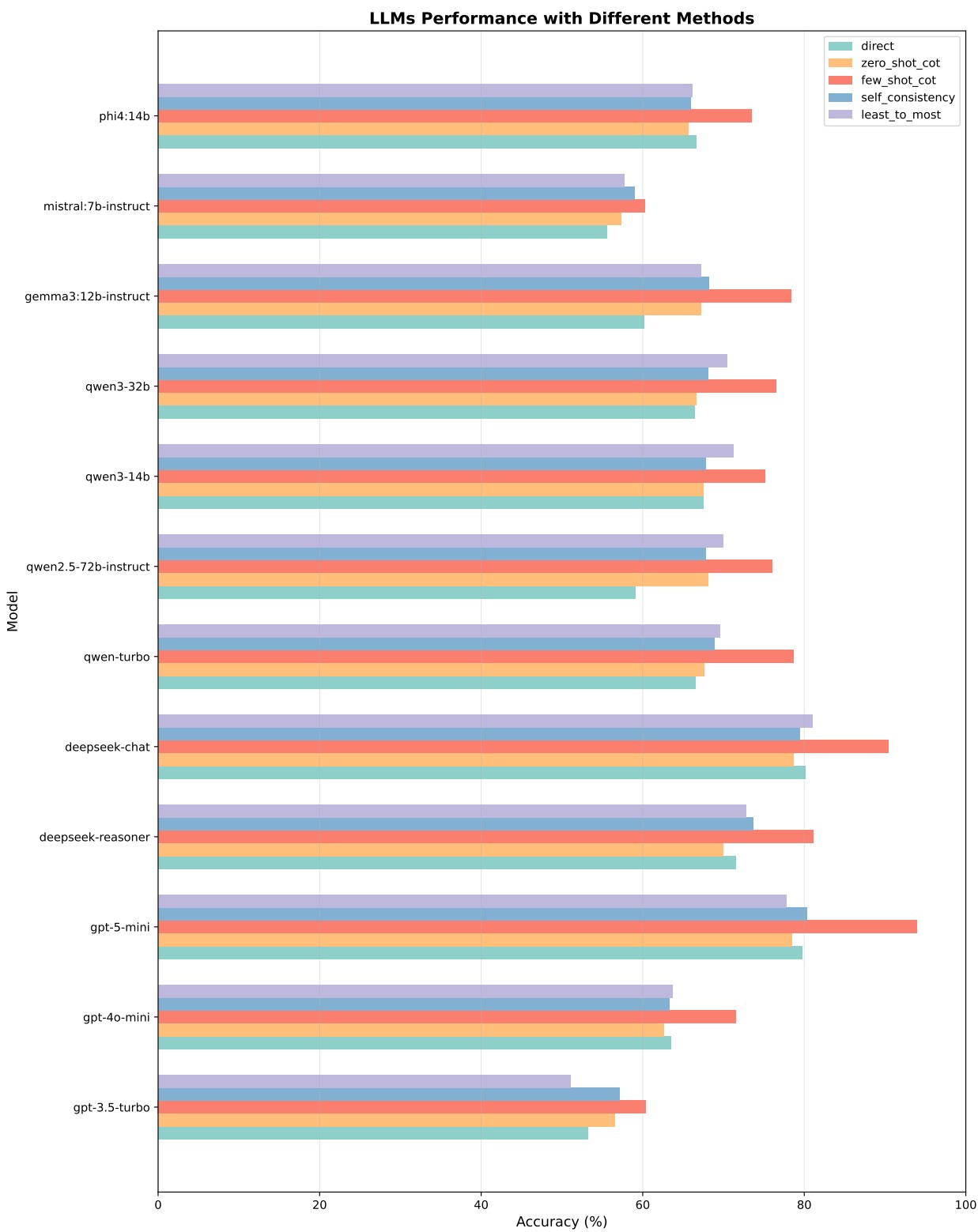

Figure 3: LLMs Performance with Different Methods.

at next state of *event 1*, which is however incorrect[5]. The second is **Context Hypothesis Detachment**,

---

[5]Two examples of *Temporal Semantics and Reasoning Misalignment* are demonstrated in Listing 4 and Listing 5 in Appendix A.2.

which indicates that LLMs largely ignore the context and only leverage information from the hypothesis to perform reasoning. A representative example is observed from the results of *Qwen-Turbo* with the *Direct Prompting* method, in which since *C1* and *C2* are identical, when asked whether both *C1* and *C2* hold, it reasons in a shortcut way without leveraging context information about the transitions of events, and directly gives the answer that *C3* is true, which however is not. We observe this happens in several models such as *Qwen-Turbo, GPT-4o-Mini, Mistral:7B-Instruct*, etc[6]. The third is **Reasoning Error Amplification**, which means the previous reasoning errors without retrospection for correcting lead to the later reasoning ignoring the errors and using the previous incorrect steps or conclusions to future reason, resulting in that the reasoning errors are amplified during the reasoning process. This is a normal and typical error when LLMs perform reasoning and we observe it in the reasoning steps of several models such as *GPT-3.5-Turbo, DeepSeek-Chat, Qwen3:14B*, etc., which is due to the methods equipped by LLMs lacks of ability for retrospection[7]. This case is not only observed in this work but also reported in prior works (Zhu et al., 2025; Feng et al., 2025; Tyen et al., 2024).

### 5.2 Impact of Increasing $m$

We select 3 LLMs, namely *DeepSeek-Reasoner, GPT-5-Mini*, and *Qwen-Turbo*, and evaluate them on the additional constructed datasets, where the number of events is fixed to $n = 2$ while the number of formula operators $m$ increases from 1 to 9, specifically $m \in \{1, 2, 3, 4, 5, 7, 9\}$. This aims to explore whether the performance of LLMs on TR challenges is stable as the number of formula operators increases and also whether the increase of the number of formula operators can introduce more complexity.

The results of the models equipped with each method are demonstrated in Figure 4. While four methods exhibit an oscillation of accuracy while $m < 7$, the *Few-Shot CoT* method maintains a stable and robust performance. Furthermore, a sudden accuracy drop is observed at $m = 3$ for the other four methods. In particular, *Direct Prompting* shows an accuracy decrease at $m = 3$ followed by an increase up to $m = 7$, indicating poorer performance on simpler problems but improved performance on harder ones, which is counter-intuitive. Thus, to further investigate the reason, we conduct a qualitative analysis. We find that when $m <= 3$, the problems are relatively simple and the LLMs do not automatically and frequently invoke explicit reasoning to arrive at answers, in which while $m < 3$, the problems could be easy for LLMs to solve even without explicit reasoning, whereas while $m = 3$, the problems become hard for the LLMs and they still fail to invoke explicit reasoning process or they put less effort in reasoning process, leading to worse performance, which is a case that especially happens frequently in *Direct Prompting*. However, while $m > 3$, the LLMs invoke reasoning more frequently and take more effort in the reasoning process, resulting in the increase of accuracy.

In addition, while $m \geq 7$, the accuracy shows an obvious decreasing trend. From $m = 7$ to $m = 9$, grouped by methods across models, accuracy decreases by 6.44% in *Direct Prompting*, 7.44% in *Zero-Shot CoT*, 3.89% in *Few-Shot CoT*, 6.22% in *Self-Consistency*, 7.11% in *Least-to-Most*, while grouped by models across methods, accuracy decreases by 6.87% in *DeepSeek-Reasoner*, 7.53% in *GPT-5-Mini*, and 4.27% in *Qwen-Turbo*. Overall, increasing the number of formula operators from 7 to 9 results in an average accuracy drop of 6.22%, with a minimum drop of 0.3% and the maximum drop of 12%. This indicates that the increase of the number of formula operators can significantly introduce more complexity to the problems and also shows that the TR ability of LLMs lacks consistency and robustness as TR problems complexity grows due to involving more formula operators.

### 5.3 Impact of increasing $n$

In addition, the selected 3 models are evaluated on another additional constructed datasets, where the number of formula operators is fixed to $m = 2$ while the number of events $n$, where $n > 1$, increases from 2 to 9, specifically $n \in \{2, 3, 4, 5, 7, 9\}$. Similar to Section 5.2, this, however, aims to explore how the TR ability of LLMs is affected as the number of events increases and also whether the increase of the number of events can lead to more complexity.

---

[6]Two examples of *Context Hypothesis Detachment* are demonstrated in Listing 6 and Listing 7 in Appendix A.2.

[7]Two examples of *Reasoning Error Amplification* are demonstrated in Listing 8 and Listing 9 in Appendix A.2.

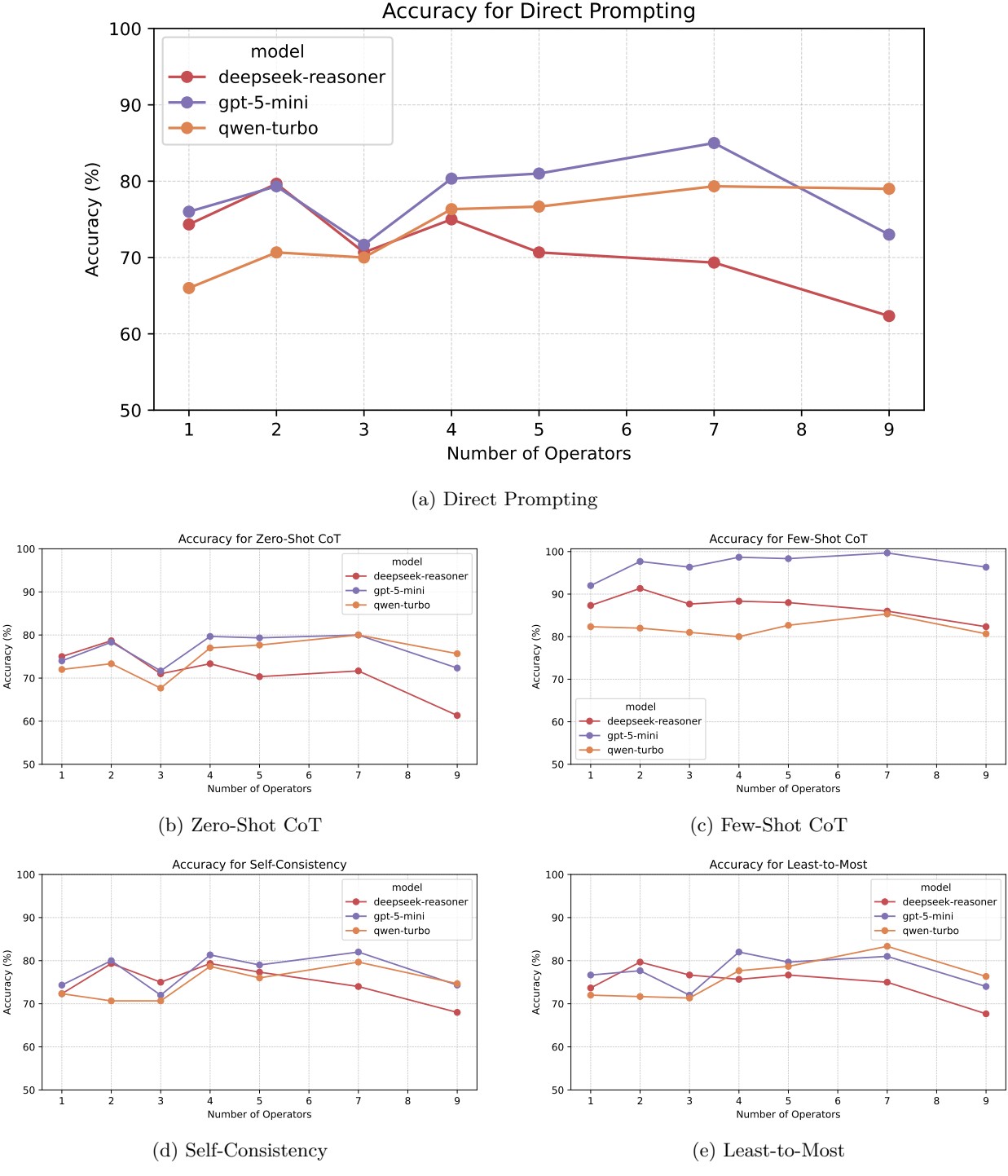

(a) Direct Prompting

(b) Zero-Shot CoT

(c) Few-Shot CoT

(d) Self-Consistency

(e) Least-to-Most

Figure 4: Accuracy for *DeepSeek-Reasoner*, *GPT-5-Mini*, and *Qwen-Turbo* equipped with each method as the number of formula operators increases.

As shown in Figure 5, across all models and methods, they show a more consistent trend of accuracy decrease as the number of events increases. From $n = 2$ to $n = 9$, grouped by methods across models, accuracy drops by 16.22% in *Direct Prompting*, 18.78% in *Zero-Shot CoT*, 19.44% in *Few-Shot CoT*, 18% in *Self-Consistency*, and 17% in *Least-to-Most*, while grouped by models across methods, accuracy decreases

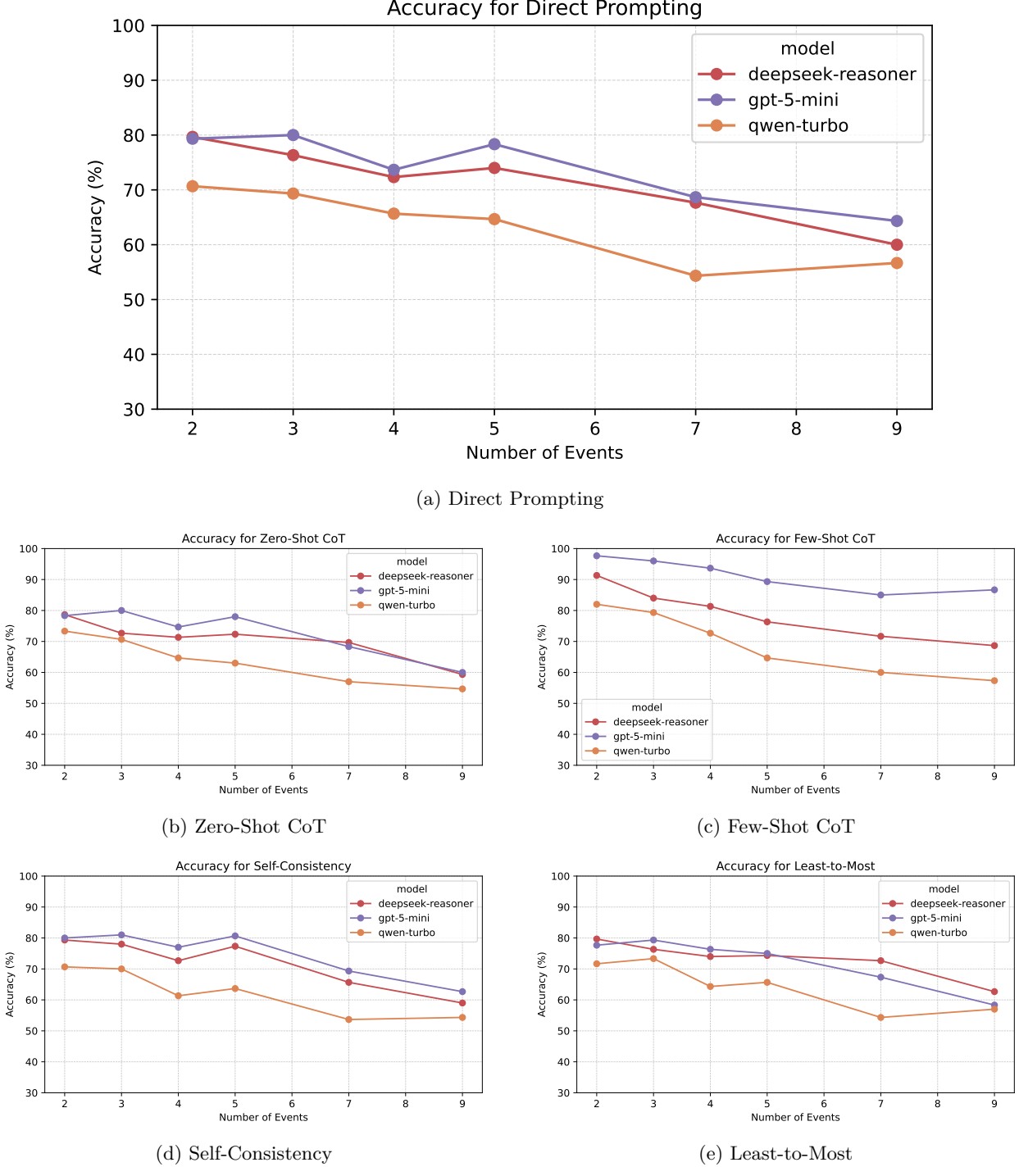

(a) Direct Prompting

(b) Zero-Shot CoT

(c) Few-Shot CoT

(d) Self-Consistency

(e) Least-to-Most

Figure 5: Accuracy for *DeepSeek-Reasoner*, *GPT-5-Mini*, and *Qwen-Turbo* equipped with each method as the number of events increases.

by 19.80% in *DeepSeek-Reasoner*, 16.20% in *GPT-5-Mini*, and 17.67% in *Qwen-Turbo*. Overall, increasing the number of events from 2 to 9 results in an average accuracy drop of 17.89%, with a minimum drop of 11% and a maximum drop of 24.67%. This decline is more pronounced than that observed when increasing the number of formula operators, indicating that the increase of the number of events can considerably lead

to more complexity of TR problems and that the TR ability of LLMs is unstable and not robust as the complexity of TR problems increases due to involving more events.

## 6 Conclusion

In this work, we evaluate TR ability of LLMs from the perspective of formal logical reasoning over temporal information and design a pipeline for TR challenges generation, by leveraging random graph generation, LTL formula, and the NuSMV model checker. Using this pipeline, we construct a dataset, LTLBench, consisting of 2000 TR challenges, and conduct intensive evaluations on 12 LLMs across 5 methods. The results demonstrate the limitations of LLMs in handling the TR problem. We further qualitatively analyze their reasoning processes and identify 3 main reasoning issues. In addition, with additional experiments, we demonstrate that their performance is unstable and not robust as the number of formula operators and events increases, and also show that the pipeline can synthesize TR challenges in various levels of complexity and size.

## Limitations

This work mainly leverages LTL to generate TR problems, of which the expression ability is limited at linear event sequences, lacking of expression ability to handle events in branching-time. Thus, a natural step of the future direction is to utilize Computation Tree Logic (CTL) or CTL* to further construct more complicated TR problems. For example, in CTL, we can use $AF\phi$ to express that for all event sequences, $\phi$ eventually holds. We believe integrating the CTL and CTL* into this pipeline could further explore the TR ability boundaries of LLMs.

In addition, this work focuses specifically on formal logical reasoning over temporal information using LTL, and therefore several temporal aspects examined in prior benchmarks fall outside the scope of LTLBench. For example, our work does not capture reasoning aspects over some temporal semantics such as *EventAtWhatTime* and *RelationDuration* evaluated in Test of Time (Fatemi et al., 2024). In addition, our work does not include reasoning aspects on temporal arithmetic such as *AddSubtract* and *Timezone* in Test of Time. Furthermore, other temporal challenges such as *Ordering* and *Frequency* mentioned in TRAM (Wang & Zhao, 2024) are not captured in this work. Conversely, the formal logical reasoning perspective over temporal information that LTLBench provides is not covered by these existing benchmarks, so LTLBench offers a complementary view rather than a replacement, and we leave the integration of these complementary temporal aspects as future work.

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

# A   Appendix

## A.1   Prompts

---

**Prompt Template 1: Direct Prompting, in which *question* is the placeholder for TR problem.**

{question}

The answer must be either 'True' or 'False'.

---

**Prompt Template 2: Zero-Shot CoT, in which *question* is the placeholder for TR problem.**

{question}

Let's think step by step.

First, let's identify what the hypothesis is asking.
Then, let's understand and examine each condition.
Let's trace and reason through the event sequences.
Finally, let's determine if the hypothesis can be satisfied.

Please reason through this problem thoroughly before answering.
The final answer must be either 'True' or 'False'.

---

**Prompt Template 3: Few-Shot CoT, in which *question* is the placeholder for TR problem.**

=== Example 1 ===

Context:
Initially, event3 happened. After event3, either event1, or event2 must happen. After event1, either event2, or event3 must happen. After event2, event3 must happen.

Hypothesis:
C1: Event1 happens or event3 happens.
C2: Event1 happens or C1 holds.
C3: C2 eventually holds.

Let's think step by step:
1. We need to check if C3 (C2 eventually holds) is true.
2. C2 is: Event1 happens or C1 holds
3. C1 is: Event1 happens or event3 happens
4. So C2 simplifies to: Event1 happens or (Event1 happens or event3 happens)
5. This is logically equivalent to: Event1 happens or event3 happens
6. The initial state has event3 = true
7. Therefore, C2 holds at the initial state (since event3 is true)
8. Since C2 holds at the initial state, C3 (C2 eventually holds) is True

Answer: True

=== Example 2 ===

Context:

---

Initially, event2 happened. After event2, event3 must happen. After event3, no other events can happen. After event1, either event2, or event3 must happen.

Hypothesis:
C1: Event2 happens and event3 happens.
C2: C1 holds in the next state.
C3: C2 eventually holds.

Let's think step by step:
1. We need to check if C3 (C2 eventually holds) is true.
2. C2 is: C1 holds in the next state, where C1 is (event2 AND event3)
3. For C1 to be true, both event2 and event3 must be true simultaneously
4. In any single state, only one event can be true at a time
5. Starting from event2, the next state must be event3
6. From event3, it stays at event3 (no other events can happen)
7. At no point can event2 and event3 be true simultaneously in the same state
8. Therefore, C1 is always false
9. If C1 is always false, then C2 (C1 in the next state) is always false
10. If C2 is always false, then C3 (C2 eventually holds) is False

Answer: False

=== Now solve this problem ===

""" + {question} + """

Let's think step by step.

First, let's identify what the hypothesis is asking.
Then, let's understand and examine each condition.
Let's trace and reason through the event sequences.
Finally, let's determine if the hypothesis can be satisfied.

Please reason through this problem thoroughly before answering.
The final answer must be either 'True' or 'False'.

---

**Prompt Template 4: Least-to-Most Question Breakdown, in which *question* is the placeholder for TR problem.**

{question}

Let me break this down into smaller, manageable parts:

1. What events are involved and what are the initial conditions?
2. What are the transition rules between events?
3. What does each sub-condition (C1, C2, C3, etc.) mean?
4. How do these conditions relate to each other?

Please answer each of these questions step by step.

Prompt Template 5: Least-to-Most Question Solving, in which *question* is the placeholder for TR problem and *breakdown_response* is the response of LLMs after prompted to breakdown a given question.

{question}

Based on the following breakdown of the problem:

{breakdown_response}

Now, let's use this understanding to solve the main problem step by step:

1. Start with the simplest conditions and work up to the complex ones
2. Trace through the event sequences
3. Check if the final hypothesis can be satisfied

The final answer must be either 'True' or 'False'.

## A.2 Failure Modes Examples

```
1  === Context ===
2
3  Initially, event1 happened. After event2, either event1, or event3 must happen. After event1
     ↪ , event3 must happen. After event3, event2 must happen. After event3, event2 must
     ↪ happen.
4
5  === Hypothesis ===
6
7  C1: Event2 happens in the next state.
8  C2: C1 eventually holds.
9  C3: Event3 happens and C2 holds.
10
11 C3 is True or False?
12
13 === Response ===
14
15 Let's analyze the problem step by step.
16
17 ### Step 1: Understanding the Context
18 - Initially, **event1** happens.
19 - After **event2**, either **event1** or **event3** must happen.
20 - After **event1**, **event3** must happen.
21 - After **event3**, **event2** must happen.
22 - After **event3**, **event2** must happen (this is repeated).
23
24 ### Step 2: Analyzing the Hypothesis
25 - **C1**: Event2 happens in the next state.
26 - **C2**: C1 eventually holds.
27 - **C3**: Event3 happens and C2 holds.
28
29 ### Step 3: Tracing the Event Sequences
30 1. **Initial State**: event1 is true.
31 2. From event1, the next event must be event3 (as per the context).
32 3. Now, we have event3 true.
33 4. After event3, event2 must happen (as per the context).
34 5. Now, we have event2 true.
35
36
37 ### Step 4: Evaluating C1, C2, and C3
38 - **C1**: Event2 happens in the next state.
39   - In the initial state, event1 is true, and the next state will be event3. After event3,
        ↪ event2 will happen. Therefore, C1 will hold true in the next state after event3.
40
41 - **C2**: C1 eventually holds.
42   - Since C1 holds true after event3, C2 is also true because C1 eventually holds.
43
44 - **C3**: Event3 happens and C2 holds.
45   - Event3 does happen (as we established in the sequence), and C2 holds true (as we
        ↪ established that C1 eventually holds). Therefore, C3 is true.
46
47 ### Conclusion
48 Since C3 is true (both conditions of C3 are satisfied), the final answer is:
49
50 **Answer: True**
```

Listing 4: Example 1 of Temporal Semantics and Reasoning Misalignment.

```
1  === Context ===
2
3  Initially, event1 happened. After event2, either event1, or event3 must happen. After event1
     ↪ , event3 must happen. After event3, event2 must happen. After event3, event2 must
     ↪ happen.
4
5  === Hypothesis ===
6
7  C1: Event2 happens in the next state.
8  C2: C1 eventually holds.
9  C3: Event3 happens and C2 holds.
10
11 C3 is True or False?
12
13 === Response ===
14
15 Step 1 - identify what C3 asks:
16 - C3 = (Event3 happens) AND (C2 holds).
17 - C2 = "C1 eventually holds."
18 - C1 = "Event2 happens in the next state" (i.e., at some state s, the immediate successor of
     ↪  s has event2).
19
20 Step 2 - use the context to build a timeline from the initial state:
21 - t0: Initially event1 happened.
22 - After event1, event3 must happen -> t1: event3.
23 - After event3, event2 must happen -> t2: event2.
24
25 Step 3 - check C1, C2, and the "Event3 happens" part:
26 - At t1, the next state (t2) has event2, so C1 holds at t1.
27 - Therefore C2 ("C1 eventually holds") is true.
28 - Event3 does occur (at t1).
29
30 Since both conjuncts of C3 are true, C3 is True.
31
32 Final answer: True
```

Listing 5: Example 2 of Temporal Semantics and Reasoning Misalignment.

```
1   === Context ===
2
3   Initially , event2 happened . After event3 , event1 must happen . After event1 , event3 must
        ↪ happen . After event2 , event1 must happen .
4
5   === Hypothesis ===
6
7   C1: Event2 always happens .
8   C2: Event2 always happens .
9   C3: C1 holds and C2 holds .
10
11  C3 is True or False ?
12
13  === Response ===
14
15  The answer is: **True**
16
17  ### Explanation :
18
19  - The hypothesis states :
20      - **C1: Event2 always happens .**
21      - **C2: Event2 always happens .**
22      - **C3: C1 holds and C2 holds .**
23
24  Since **C1** and **C2** are ** identical statements **, both asserting that "Event2 always
        ↪ happens ," then **C3** ( which says "C1 holds and C2 holds ") is logically ** true **
        ↪ because both conditions are the same and therefore both true .
25
26  Thus , **C3 is True **.
```

Listing 6: Example 1 of Context Hypothesis Detachment.

```
1   === Context ===
2
3   Initially , event2 happened . After event3 , event1 must happen . After event1 , event3 must
        ↪ happen . After event2 , event1 must happen .
4
5   === Hypothesis ===
6
7   C1: Event2 always happens .
8   C2: Event2 always happens .
9   C3: C1 holds and C2 holds .
10
11  C3 is True or False ?
12
13  === Response ===
14
15  **Answer :** True
16
17  **Explanation :**
18
19  - C1 says: "Event2 always happens ."
20  - C2 says: "Event2 always happens ."
21  - C3 says: "C1 holds and C2 holds ."
22
23  Since both C1 and C2 are identical and assert the same thing ( that Event2 always happens ),
        ↪ and both are true , then their conjunction (C3) is also ** true **.
24
25  Thus , **C3 is True **.
```

Listing 7: Example 2 of Context Hypothesis Detachment.

```
1  === Context ===
2
3  Initially, event3 happened. After event2, either event1, or event3 must happen. After event1
       ↪ , no other events can happen. After event3, event1 must happen.
4
5  === Hypothesis ===
6
7  C1: Event2 always happens.
8  C2: C1 holds and event1 happens.
9  C3: C2 holds and event3 happens.
10
11 C3 is True or False?
12
13 === Response ===
14
15 The hypothesis states that event2 always happens, event1 happens after event2, and event3
       ↪ happens after event1.
16
17 Let's examine each condition:
18 - C1: Event2 always happens. This means that event2 is a necessary condition for the
       ↪ sequence of events to occur.
19 - C2: C1 holds and event1 happens. This means that event1 follows event2, as stated in the
       ↪ hypothesis.
20 - C3: C2 holds and event3 happens. This means that event3 follows event1, as stated in the
       ↪ hypothesis.
21
22 Now, let's trace the event sequences:
23 1. Event2 happens (C1 is satisfied).
24 2. Event1 happens after event2 (C2 is satisfied).
25 3. Event3 happens after event1 (C3 is satisfied).
26
27 Based on the given conditions and the sequence of events, C3 is True. Event3 does happen
       ↪ after event1 in this scenario.
```

Listing 8: Example 1 of Reasoning Error Amplification.

```
1  === Context ===
2
3  Initially, event3 happened. After event2, no other events can happen. After event1, no other
       ↪   events can happen. After event3, event2 must happen.
4
5  === Hypothesis ===
6
7  C1: Event3 does not happen.
8  C2: Event3 does not happen.
9  C3: If C1 holds, then C2 holds.
10
11 C3 is True or False?
12
13 === Response ===
14
15 First, let's identify what the hypothesis is asking. The hypothesis states that if event3
       ↪   does not happen (C1), then event2 does not happen (C2). We need to determine if this
       ↪   conditional statement is true or false.
16
17 Now, let's examine each condition:
18 - C1: Event3 does not happen.
19 - C2: Event2 does not happen.
20
21 Next, let's trace and reason through the event sequences:
22 1. Initially, event3 happened.
23 2. After event3, event2 must happen.
24 3. After event2, no other events can happen.
25 4. After event1, no other events can happen.
26
27 Given the sequence of events, if event3 does not happen (C1), then event2 cannot happen,
       ↪   which contradicts the given sequence of events. Therefore, C3 (if C1 holds, then C2
       ↪   holds) is false.
28
29 Therefore, the final answer is False.
```

Listing 9: Example 2 of Reasoning Error Amplification.

### A.3 Pseudocode

---

**Algorithm 1** Generate LTL Formulas

---

1: **procedure** GENERATELTLFORMULAS($states, formulaLength, formulaCount$)
  **Input:** $states$ - an array of atomic states representing events
  **Input:** $formulaLength$ - the number of operators
  **Input:** $formulaCount$ - the number of formulas to generate
  **Output:** an array of LTL formulas
2:    $unaryOperators \leftarrow [X, G, F, !]$
3:    $binaryOperators \leftarrow [\&, |, \rightarrow]$
4:    $operators \leftarrow unaryOperators + binaryOperators$
5:    $B \leftarrow [[\ ]$ of size $formulaLength + 1\ ]$
6:    $B[0] \leftarrow states$
7:    $formulas \leftarrow [\ ]$
8:    **for** $i \leftarrow 1$ **to** $formulaCount$ **do**
9:        **for** $j \leftarrow 1$ **to** $formulaLength$ **do**
10:          $x \leftarrow$ randomly choose an operator from $operators$
11:          **if** $x \in unaryOperators$ **then**
12:              $y \leftarrow$ randomly sample a formula from $B[j-1]$
13:              $newFormula \leftarrow [x, y]$
14:          **else**
15:              $s \leftarrow$ randomly sample a integer from $[0, j)$
16:              $y1 \leftarrow$ randomly sample a formula from $B[s]$
17:              $y2 \leftarrow$ randomly sample a formula from $B[j-1-s]$
18:              $newFormula \leftarrow [y1, x, y2]$
19:          **end if**
20:          append $newFormula$ to $B[j]$
21:        **end for**
22:        $formula \leftarrow B[formulaLength][-1]$
23:        append $formula$ to $formulas$
24:    **end for**
25:    **return** $formulas$
26: **end procedure**

---

---

**Algorithm 2** Template-Based Natural Language Context Generation

---

1: **procedure** GENERATENLCONTEXT($G, e_{init}$)
    **Input:** $G = (V, E)$ – the random directed graph
    **Input:** $e_{init} \in V$ – the initial event
    **Output:** the context represented in natural language
2:     $text \leftarrow$ "Initially, $e_{init}$ happened."
3:     **for** each event $e_i \in V$ **do**
4:         $S_i \leftarrow \{e_j \mid (e_i, e_j) \in E \wedge e_j \neq e_i\}$
5:         **if** $S_i = \emptyset$ **then**
6:             $s \leftarrow$ "After $e_i$, no other events can happen."
7:         **else if** $|S_i| = 1$ and $(e_i, e_i) \notin E$ **then**
8:             Let $e_j \in S_i$
9:             $s \leftarrow$ "After $e_i$, $e_j$ must happen."
10:        **else**
11:            $s \leftarrow$ "After $e_i$, either JOIN($S_i$) must happen."
12:        **end if**
13:        $text \leftarrow text \parallel s$
14:     **end for**
15:     **return** $text$
16: **end procedure**

---

---

**Algorithm 3** Template-Based Natural Language Hypothesis Generation

---

1: **procedure** GENERATENLHYPOTHESIS($\phi, \mathcal{B}$)
    **Input:** $\phi$ – an LTL formula (a nested list)
    **Input:** $\mathcal{B}$ – the set of base (atomic) states, i.e., events
    **Output:** a list of labeled clauses $L$ and the root label $C_{root}$
2:    $L \leftarrow [\,], \; k \leftarrow 0$
3:    $C_{root} \leftarrow$ TRANSLATE($\phi, \mathcal{B}, L, k$)
4:    **return** $L, C_{root}$
5: **end procedure**

6: **procedure** TRANSLATE($\phi, \mathcal{B}, L, k$)
7:    **for** each $i \in \{1, \ldots, |\phi|\}$ **do**
8:        **if** $\phi[i]$ is a list **then**
9:            $\phi[i] \leftarrow$ TRANSLATE($\phi[i], \mathcal{B}, L, k$)
10:       **end if**
11:    **end for**
12:    $k \leftarrow k + 1$
13:    **if** $|\phi| = 2$ **then**                                           ▷ Unary operator
14:        $op \leftarrow \phi[1], \; u \leftarrow \phi[2]$
15:        $v \leftarrow$ "happens" **if** $u \in \mathcal{B}$ **else** "holds"
16:        **if** $op = X$ **then**
17:            $clause \leftarrow$ "$\{u\}$ $\{v\}$ in the next state"
18:        **else if** $op = G$ **then**
19:            $clause \leftarrow$ "$\{u\}$ always $\{v\}$"
20:        **else if** $op = F$ **then**
21:            $clause \leftarrow$ "$\{u\}$ eventually $\{v\}$"
22:        **else if** $op = \neg$ **then**
23:            $v' \leftarrow$ "happen" **if** $u \in \mathcal{B}$ **else** "hold"
24:            $clause \leftarrow$ "$\{u\}$ does not $\{v'\}$"
25:        **end if**
26:    **else if** $|\phi| = 3$ **then**                                ▷ Binary operator
27:        $\ell \leftarrow \phi[1], \; op \leftarrow \phi[2], \; r \leftarrow \phi[3]$
28:        $\ell \leftarrow$ "$\{\ell\}$ happens" **if** $\ell \in \mathcal{B}$ **else** "$\{\ell\}$ holds"
29:        $r \leftarrow$ "$\{r\}$ happens" **if** $r \in \mathcal{B}$ **else** "$\{r\}$ holds"
30:        **if** $op = \wedge$ **then**
31:            $clause \leftarrow$ "$\{\ell\}$ and $\{r\}$"
32:        **else if** $op = \vee$ **then**
33:            $clause \leftarrow$ "$\{\ell\}$ or $\{r\}$"
34:        **else if** $op = \rightarrow$ **then**
35:            $clause \leftarrow$ "If $\{\ell\}$, then $\{r\}$"
36:        **end if**
37:    **end if**
38:    Capitalize the first character of *clause*
39:    Append "$C_{\{k\}}$: $\{clause\}$." to $L$
40:    **return** $C_k$
41: **end procedure**

---

## A.4   Dataset Characterization

To demonstrate the characterization of LTLBench, we further provide 5 statistics on this main dataset, i.e., LTLBench with the number of events = 3, the number of formula operators = 3, and 2000 instances. Each statistic is explained as follows:

- **Directed Graphs Distinction**: We count both the number of raw distinct transition graphs, which means that two graphs are counted as distinct if they differ in any edge, and the number of distinct isomorphism classes, which means that two graphs are counted as distinct if they differ in structure regardless of the specific event identifiers. The number of raw distinct transition graphs is 64, with mean = 31.25 and std = 5.37 instances per raw distinct transition graph. The number of distinct isomorphism classes is 16, with mean = 125 and std = 59.72 instances per distinct isomorphism class.

- **Formula Structure Distinction**: For each formula, we replace all event identifiers with a single placeholder symbol so that only the operator structure is retained, and then deduplicate. This yields 596 distinct formula structures out of 2000 instances, with mean = 3.36 and std = 3.07 instances per distinct formula structure.

- **Graph Edge Density**: We count the number of directed edges in each graph, with mean edge density = 3 and std = 1.23, matching the edge density expectation of ER model $G(n, p)$, i.e., $p \cdot n(n-1) = 3$ with $p = 0.5$ and $n = 3$.

- **Unary and Binary Operators per Formula**: For each formula, we count the number of unary operators (e.g., $X$) and binary operators (e.g., logical AND "&"). For unary operators, mean = 1.73 and std = 0.9 operators per instance. For binary operators, mean = 1.27 and std = 0.9 operators per instance.

- **Natural-Language Problem Length**: For each problem instance, we measure the character length of the concatenation of the context and hypothesis represented in natural language, with mean = 307.17 and std = 24.53 characters per instance.

