# OpenReview forum: "LTLBench: Towards Benchmarks for Evaluating Temporal Reasoning in Large Language Models"
_TMLR — Decision pending for TMLR_

### Review · Reviewer_xPFt · 2026-04-28

**Summary Of Contributions:**

This work focus on benchmarking the temporal reasoning capabilities of LLMs.
In particular, the authors propose an alternative way to evaluating TR ability by using Linear Temporal Logic, which is a formal logic used for studying the sequences of events over time.
Real world events can also be formulated as LTL tasks.
By developing a LTL pipeline, the authors constructed a dataset called LTLBench consisting of 2000 TR challenges while benchmarking 12 LLM models with 5 prompting techniques.
Since the previous submission, the authors have also added substantial empirical changes to facilitate more robust evaluation.



**Strengths:**
1. The use of LTL logic provides a nice framework to evaluate the temporal reasoning capabilities of LLMs and this is also a timely topic.
2. The authors conducted a comprehensive benchmark of various LLM models on the proposed LTLBench dataset.
3. A diversity of prompting techniques are included showing that few-shot CoT is the most effetive. The authors also analyzed three reasoning failures from benchmarking the LLMs.


**Weaknesses:**
1. some figures are pixelated, please make sure to use pdf for figures. For example, Figure 4(a), Figure 5(a)
2. lack exploration of choice of the random graph generator and its impact on benchmarking results
3. The paper writing can be improved. For example, the three main issues of reasoning failures are not easy to find in text and should be highlighted. In addition, in the appendix, more example failure cases and LLM outputs should be included for better understanding of their output.

**Audience:**

Yes

**Audience Explanation:**

The topic is of interest to the TMLR audience. The temporal reasoning capability of LLMs are required in increasingly more tasks such as web search, QA and summary. Having a principled benchmark to test the model's TR capability is important.

**Broader Impact Concerns:**

There are no ethical concerns in my knowledge for this work.

**Claims And Evidence:**

Yes

**Claims Explanation:**

Yes, the authors conducted extensive experiments with 12 LLM models and 5 prompting techniques to empirically validate the benchmark and report findings. Details are found in Section 5.

**Requested Changes:**

I would request the following changes.

Crucial for acceptance:
- Figure and diagram updates, all figures should be reported with pdf, please include such in your revision
- better explanation of which random graph generator family you have used. I assume it is the ER graph (Erdős–Rényi model) , please make it more clear in the paper.
- clearly highlight the reasoning failures in text, provide more detailed failure examples from a selection of models in the appendix.

Strengthen the work:
- Exploring alternative random graph generators and its impact on the task difficulty. The ER graph is often considered the easiest generator, other families would include the Barabási–Albert model, Stochastic block model. It might be interesting to investigate star graphs as well where a given event has many in degree / out degree while others have degree 1.
- Explain more about why 2000 TR challenges are used, showing if using less or more would change the difficulty by how much
- Provide a clean and easy pipeline for the public to generate more TR challenges if needed.

---

> ### Author Response · Authors · 2026-05-25
>
> Thank you for your detailed comments. We address each point below and will make the corresponding revisions to the paper.
>
> **To Comment 1**: *“Figure and diagram updates, all figures should be reported with pdf, please include such in your revision.”*
>
> We agree. In the revised version, we will ensure the attached figures in PDF format.
>
> **To Comment 2**: *“Better explanation of which random graph generator family you have used. I assume it is the ER graph (Erdős–Rényi model) , please make it more clear in the paper.”*
>
> Our pipeline uses the Erdős–Rényi random directed graph model, i.e $G(n,p)$, in which the edge probability $p=0.5$. We will explicitly mention this in Section 3.1.
>
> **To Comment 3**: *“Clearly highlight the reasoning failures in text, provide more detailed failure examples from a selection of models in the appendix..”*
>
> Thanks for pointing this out. We will not only highlight the three main reasoning failures in text but will also highlight where the reasoning failures happen in the example in Appendix.

---

> > ### Comment · Reviewer_xPFt · 2026-05-30
> > **Official Comment by Reviewer xPFt**
> >
> > Thank you for your response, most of my concerns have been addressed.
> >
> > Have you considered the points I made for Strengthening the work: For example, the impact of using more of less TR challenge and its impact of the difficulty of the task?

---

> > > ### Author Response · Authors · 2026-06-01
> > >
> > > Thank you for the follow-up.
> > >
> > > **To Comment 1 for Strengthening**: *"Exploring alternative random graph generators and its impact on the task difficulty. The ER graph is often considered the easiest generator, other families would include the Barabási–Albert model, Stochastic block model. It might be interesting to investigate star graphs as well where a given event has many in degree / out degree while others have degree 1."*
> > >
> > > We agree that this is an interesting direction. The current version of LTLBench focuses more on the side of formal logical reasoning over temporal information, and we do not impose any particular structural requirement on the transitions between events, nor do we presuppose any prior structure.  We adopt the ER random graph model because it is typically the simplest random graph model and has no structural prior beyond the edge probability, which naturally makes it the most suitable choice for this version of our pipeline. That said, we also agree that extending the pipeline to alternative generators is indeed meaningful, particularly when targeting specific TR tasks whose event transitions are expected to follow a presupposed prior structure. We believe such an exploration is most valuable when conducted in combination with a concrete downstream task in which a particular prior graph structure carries real significance, while the current pipeline serves as a neutral foundation that does not impose any such prior. Adapting the graph generation model to a corresponding family (e.g., Barabási–Albert graph model, or Stochastic Block Model) is therefore a natural direction for future work targeting structure specific TR problems.
> > >
> > > **To Comment 2 for Strengthening**: *"Explain more about why 2000 TR challenges are used, showing if using less or more would change the difficulty by how much."*
> > >
> > > We consider 2000 to be an appropriate size to evaluate the performance of LLMs, providing adequate coverage of problem diversity. We have further analyzed these 2000 challenges from 5 perspectives, i.e., *Directed Graphs Distinction*, *Formula Structure Distinction*, *Graph Edge Density*, *Unary / Binary Operators per Formula*, and *Natural-Language Problem Length*, and reported the corresponding statistics in our response to Reviewer 644A.
> > >
> > > **To Comment 3 for Strengthening**: *"Provide a clean and easy pipeline for the public to generate more TR challenges if needed."*
> > >
> > > Our code is open-sourced as shown in Footnote 1 at Page 1, and the link is anonymized for now in accordance with the double-blind review policy. The released code includes clear instructions for use, allowing users to easily modify the pipeline or directly use it to generate more complex and diverse TR challenges. Once the paper is accepted, we will de-anonymize the repository and additionally release a HuggingFace dataset link to further facilitate community use.

---

### Review · Reviewer_644A · 2026-05-07

**Summary Of Contributions:**

The paper introduces LTLBench, a synthetic benchmark and generation pipeline for evaluating temporal reasoning in LLMs. The pipeline generates random directed graphs, samples LTL formulas, uses NuSMV for ground-truth labels, and converts problems into natural language prompts. The revised version evaluates 12 LLMs across 5 prompting/reasoning methods, adds experiments on the effect of increasing events and operators, and includes a qualitative analysis of reasoning failures.

Some limitations remain regarding benchmark representativeness and the motivation for random graph generation, but the revision addresses several of my earlier concerns.

**Audience:**

Yes

**Audience Explanation:**

Yes. The paper is relevant to researchers interested in LLM reasoning and temporal reasoning.

**Broader Impact Concerns:**

I do not have specific broader impact concerns.

**Claims And Evidence:**

Yes

**Claims Explanation:**

The revised version provides stronger evidence than the previous submission. The authors now evaluate a broader and more up-to-date set of models, compare multiple prompting/reasoning methods. These additions make the empirical evaluation more convincing and improve the diagnostic value of the benchmark.

Some aspects could still be strengthened, especially the justification of the random graph generation procedure and a more systematic characterization of dataset diversity and difficulty. However, the paper now frames the benchmark more clearly as an LTL-based perspective on temporal reasoning, and the evidence is sufficient to support the main claims under this framing.

**Requested Changes:**

- Better motivate the random graph generation procedure and discuss how graph properties may affect task difficulty.
- Add more characterization of dataset diversity and difficulty, beyond only the number of events and formula operators.
- Broaden the limitations discussion on what kinds of temporal reasoning are not captured by the current LTL formulation.

---

> ### Author Response · Authors · 2026-05-25
>
> Thank you for constructive suggestions. We respond each comments as follows:
>
> **To Comment 1**: *“Better motivate the random graph generation procedure and discuss how graph properties may affect task difficulty.”*
>
> We will revise Section 3.1 to explicitly explain that our pipelines use Erdős–Rényi random directed graph model, with edge probability 0.5. We will explain why the ER graph model is more suitable than others for the current version of our pipeline.
>
> **To Comment 2**: *“Add more characterization of dataset diversity and difficulty, beyond only the number of events and formula operators.”*
>
> We will run a script to discover additional characteristics, such as the distribution of LTL operator types over problems, and explain it in Appendix.
>
> **To Comment 3**: *“Broaden the limitations discussion on what kinds of temporal reasoning are not captured by the current LTL formulation.”*
>
> We will expand the Limitations section to discuss aspects of temporal reasoning not captured by LTL, including (1) branching-time properties expressible in CTL/CTL*, (2) temporal aspects that are supported in other benchmarks but are not currently included in this work like temporal arithmetic.

---

> > ### Comment · Reviewer_644A · 2026-05-29
> >
> > Thank you for agreeing to incorporate these points in the revised version.
> > But to give a final recommendation, I need a bit more detail on the following:
> >
> > 1. Could you clarify why the Erdős–Rényi model is the most appropriate choice for your pipeline, and specifically why an edge probability of 0.5 was selected?
> >
> > 2. Could you provide the output or summary statistics from the script used to characterize dataset diversity and difficulty?
> >
> > 3. Could you briefly indicate what text you plan to add to the Limitations section regarding temporal reasoning aspects not captured by the current LTL formulation?

---

> ### Author Response · Authors · 2026-05-29
>
> Thank you for the follow-up. We respond to each point as follows:
>
> **To Comment 1**: *"Could you clarify why the Erdős–Rényi model is the most appropriate choice for your pipeline, and specifically why an edge probability of 0.5 was selected?"*
>
> We choose the ER random graph model because we do not impose any particular structural requirement on the transitions between events, nor do we presuppose any prior structure. The ER model is typically the simplest random-graph model and has no structural prior beyond the edge probability, which naturally makes it the most suitable choice for our pipeline. Within ER, $p = 0.5$ is the most unbiased choice, since each of the $n(n-1)$ candidate directed edges is independently present or absent with equal probability, with no preference for sparser or denser transition structures. Of course, one could adopt other random graph models or stronger prior structures. Nevertheless, since the focus of this work is on formal logical reasoning over temporal information as we discussed in the paper, the ER model is the simpler and more natural choice.
>
> **To Comment 2**: *"Could you provide the output or summary statistics from the script used to characterize dataset diversity and difficulty?"*
>
> We have run the script on the main set (i.e, n = 3, m = 3, 2000 instances, with label balance 50.0% True and 50.0% False). We characterize the dataset by 5 statistics:
>
> 1. **Directed Graphs Distinction**: We count both the raw distinct transition graphs (by labeled edge set, i.e., graphs with any difference in their edges are counted as distinct) and the distinct isomorphism classes (structurally distinct graphs after relabeling event identifiers, computed by networkx.is_isomorphic). The raw distinct graphs is 64, with $mean = 31.25$, $std = 5.37$ instances per labeled graph. The distinct isomorphism classes is 16, with $mean = 125$, $std = 59.72$ instances per class.
> 2. **Formula Structure Distinction**: For each formula we replace all event identifiers with a single placeholder symbol so that only the operator structure is retained, then deduplicate. Distinct formula structures are 596 out of 2000, with $mean = 3.36$, $std = 3.07$ instances per structure.
> 3. **Graph Edge Density**: We compute the number of edges per graph, with $mean = 3$, $std = 1.23$, matching the ER model $p \times n(n-1) = 3$ (where $p = 0.5$ is edge probability we use).
> 4. **Unary / Binary Operators per Formula**: For each formula, we count the total number of unary operators (e.g, X) and binary operators (e.g. logical AND "&"). For unary, it is mean = 1.73, std = 0.9, while for binary, it is mean = 1.27, std = 0.9.
> 5. **Natural-Language Problem Length**: We measure the character length of the natural-language problem (context + hypothesis), with mean = 307.17, std = 24.53.
>
> **To Comment 3**: *"Could you briefly indicate what text you plan to add to the Limitations section regarding temporal reasoning aspects not captured by the current LTL formulation?"*
>
> In addition to the existing discussion of CTL/CTL* in the current Limitations section, we plan to add the following paragraph:
>
> *In addition, this work focuses specifically on formal logical reasoning over temporal information using LTL, and therefore several temporal aspects examined in prior benchmarks fall outside the scope of LTLBench. For example, our work does not capture reasoning aspectes over some temporal semantics such as EventAtWhatTime and RelationDuration evaluated in Test of Time (Fatemi et al., 2024). In addition, our work does not include reasoning aspects on temporal arithmetic such as AddSubtract and Timezone in Test of Time. Furthermore, other temporal challenges such as Ordering and Frequency mentioned in TRAM (Wang & Zhao, 2024) are not captured in this work. Conversely, the formal logical reasoning perspective over temporal information that LTLBench provides is not covered by these existing benchmarks, so LTLBench offers a complementary view rather than a replacement, and we leave the integration of these complementary temporal aspects as future work.*

---

### Review · Reviewer_HTm7 · 2026-05-14

**Summary Of Contributions:**

The paper introduces LTLBench, an automated framework for evaluating Temporal Reasoning (TR) in LLMs using Linear Temporal Logic (LTL). The authors developed a pipeline to synthesize 2000 challenges by converting random directed graphs and LTL formulas into natural language, verified by the NuSMV model checker. The study benchmarks 12 LLMs across 5 prompting methods and identifies three primary failure modes.

### Strengths:
- Formal Rigor: Use of NuSMV ensures mathematically certain ground truth labels.
- Scalability: The automated pipeline facilitates the creation of complex, varied TR tasks.

### Weaknesses:
- Logical Ambiguity: Certain descriptions of event transitions are logically contradictory or confusing.
- Statistical Depth: The current results lack reporting on variance and the number of trials per instance, making it difficult to assess the stability of the findings.
- Space Efficiency: The manuscript utilizes significant space for figures (e.g., Figures 4 and 5) that could be more concisely presented and no additional information between Figure 3 and Table 1.

**Audience:**

Yes

**Audience Explanation:**

The TMLR audience would benefit from a formal logic-based benchmark for TR, provided the methodological ambiguities are resolved.

**Claims And Evidence:**

No

**Claims Explanation:**

While the quantitative results are extensive, several areas lack the clarity required for TMLR:

- Logical Inconsistency: On page 3, the authors state: "The case that event1 points to event3 and event3 also points to event1 means that event1 must happen after event3 and event3 must happen after event1". This implies a temporal loop or simultaneity that is not clearly explained in a linear logic context.

- Lack of Statistical Confidence: Table 1 reports single accuracy percentages without standard deviation or variance. It is unclear how many trials were run for each single instance, which is necessary to determine if the performance gaps are statistically significant or mere noise.

**Requested Changes:**

### Critical Changes (Required for acceptance):
- Clarify Event Loop Logic: Explain the logic on page 3 where event1 and event3 "must happen" after one another. If this represents a cycle (1→3→1), the authors must clarify how this is represented in the natural language prompt and handled by the NuSMV model checker.
- Statistical Reporting: How many trials of each single instance is run for evaluation? Provide the number of trials per instance and report the standard deviation/variance for the results in Table 1. This is essential to confirm the statistical confidence of the model comparisons.
- Methodological Evidence for Reasoning: Elaborate on the claim that for $m\leq3$, models "do not automatically and frequently invoke explicit reasoning". How was this "confirmation" performed? (e.g., Did you analyze the average number of "thought" tokens, or use a specific qualitative rubric?).
- Clarify Self-Consistency Comparisons: In Section 5.1, the authors state Self-Consistency fails to improve high-performing models. Clarify if this comparison is against Zero-Shot CoT (the base method SC extends ) or Few-Shot CoT. If the latter, acknowledge that they are distinct methods rather than direct improvements of one over the other (I thought in self-consistency, you do not provide the examples).
- Address anomalous performance drops: Specifically discuss why Qwen2.5:72B-Instruct performance decreased from 68.10% (Zero-Shot CoT) to 67.75% (Self-Consistency) in Table 1. Typically, Self-Consistency is expected to improve performance; this anomaly requires a brief qualitative explanation.

- Ablation studies on different temperatures. It may be possible that the experimental results will highly depend on those parameters.

Strengthening Changes (Recommended):
- Space Efficiency and Conciseness: Figures 4 and 5 occupy substantial space. Consider merging these into a single multi-panel figure or using a more space-efficient visualization to allow for more informative text analysis.

---

> ### Author Response · Authors · 2026-05-25
>
> Thank you for the detailed comments. We respond and address each point as follows:
>
> **To Comment 1**: *“Clarify Event Loop Logic: Explain the logic on page 3 where event1 and event3 "must happen" after one another. If this represents a cycle (1→3→1), the authors must clarify how this is represented in the natural language prompt and handled by the NuSMV model checker.”*
>
> The directed graph defines a transition relation between event states in the Kripke structure, not a chronological "before/after" ordering. When event1 → event3 and event3 → event1 both exist, it means: from the state in which event1 is the current event, the next event will be event3, and symmetrically from event3 the next event will be event1. This works as a state machine and this is supported by NuSMV. In addition, to clarify clearly, we will include two pseudocodes to describe how we exactly translate these event transitions and hypothesis to natural language.
>
> **To Comment 2**: *“Statistical Reporting: How many trials of each single instance is run for evaluation? Provide the number of trials per instance and report the standard deviation/variance for the results in Table 1. This is essential to confirm the statistical confidence of the model comparisons.”*
>
> We would like to clarify our experimental settings, which we believe addresses the reviewer's concern. For Direct Prompting, Zero-Shot CoT, Few-Shot CoT, and Least-to-Most, we set temperature = 0 for all models except GPT-5-Mini whose temperature is not settable as noted in footnote 3 at page 6. At temperature = 0 the LLM output is deterministic, so a single pass over the 2000 instances benchmark yields the exact accuracy under that prompt and repeated runs would return identical results, and reporting a standard deviation across them is therefore not meaningful. When comparing models’ performance, the current results can already give a stable and confident comparison. Even for model like GPT-5-Mini for which we can not set temperature, when comparing its performance with others, for example, GPT-4o-Mini and Qwen-Turbo, GPT-5-Mini surpasses them across all methods. With evaluated on 2000 challenges across 5 methods, the consistently substantial performance gap between GPT-5-Mini and others is very hard to be attributed to stochasticity.
>
> **To Comment 3**: *“Methodological Evidence for Reasoning: Elaborate on the claim that for m<=3, models "do not automatically and frequently invoke explicit reasoning". How was this "confirmation" performed? (e.g., Did you analyze the average number of "thought" tokens, or use a specific qualitative rubric?).”*
>
> Quantitatively analyzing the average number of “thought” tokens can not exclude whether they are still reasoning in an explicit way (e.g., a short response may still contain explicit reasoning, while a long one may merely restate the problem). Our findings are therefore based on qualitative analysis through manual inspection of model responses,  which allows us to directly judge whether explicit reasoning steps are present.
>
> **To Comment 4**: *“Clarify Self-Consistency Comparisons: In Section 5.1, the authors state Self-Consistency fails to improve high-performing models. Clarify if this comparison is against Zero-Shot CoT (the base method SC extends ) or Few-Shot CoT. If the latter, acknowledge that they are distinct methods rather than direct improvements of one over the other (I thought in self-consistency, you do not provide the examples).”*
>
> This comparison is against Zero-Shot CoT. In Page 6, we mentioned “Self-Consistency using the same prompt template as Zero-Shot CoT”.
>
> **To Comment 5**: *“Address anomalous performance drops: Specifically discuss why Qwen2.5:72B-Instruct performance decreased from 68.10% (Zero-Shot CoT) to 67.75% (Self-Consistency) in Table 1. Typically, Self-Consistency is expected to improve performance; this anomaly requires a brief qualitative explanation.”*
>
> This drop is around 0.35%. With such a nuanced drop, it cannot be characterized as an anomalous performance drop. For comparison, Qwen3:14B, shows a 0.35% improvement from 67.45% (Zero-Shot CoT) to 67.80% (Self-Consistency). Such small fluctuations in both directions are expected due to temperature of SC is not 0 and do not constitute anomalies.

---

> ### Author Response · Authors · 2026-05-25
>
> **To Comment 6**: *“Ablation studies on different temperatures. It may be possible that the experimental results will highly depend on those parameters.”*
>
> We respectfully note that our experimental settings already demonstrate the possibility of temperature. As clarified in our response to Comment 2, four of the five methods (Direct Prompting, Zero-Shot CoT, Few-Shot CoT, and Least-to-Most) are evaluated with temperature=0, which yields deterministic outputs and removes temperature as a variable in the comparison and could already demonstrate how current models perform in this benchmark. In addition, Self-Consistency requires temperature > 0 by construction in order to generate diverse reasoning paths and we use temperature = 0.7, a commonly adopted value in prior works, which can also demonstrate how LLMs perform if temperature is not 0. To perform more ablation experiments on temperature would therefore not bring additional insights or very marginally, but incurs prohibitive costs. In addition, to the best of our knowledge, we are not aware of benchmark papers that perform ablation on temperature, and temperature ablations are not standard practice in benchmark papers.

---

### Comment · Action_Editor_fu98 · 2026-07-03
**Please submit a revision of the paper with all the changes mentioned in the discussion with the reviewers.**

Dear authors,

It seems you have not uploaded the revised version of the paper. For me to take my final decision, I need to see the final version of the paper. Please incorporate all the elements you mentioned you would add in the discussion with the authors and submit it as a revision of this submission here on openreview. I would kindly ask you to use colour for the changes to make it easier for me to double check.

Thanks in advance,

your AC

---

> ### Author Response · Authors · 2026-07-03
>
> Thank you for pointing this out and for the reminder. We will upload the revised version within the next few days.

---

> ### Author Response · Authors · 2026-07-05
> **Revision is uploaded**
>
> Dear Action Editor,
>
> We have uploaded the revised version. **All changes are highlighted in a red background**. The changes are summarized as follows:
> 1. **Description of Random Graph Generation**: This change is raised by (1) Reviewer 644A discussed in *To Comment 1*, (2) Reviewer xPFt discussed in *To Comment 2*. We revised it in Section 3.1 at page 3.
> 2. **Pseudocodes for Context and Hypotheses Natural Language Translation**: This change is raised by Reviewer HTm7 discussed in *To Comment 1*. We added three algorithms as Algorithms 1-3 in Appendix A.3 at pages 26-28, with references in Section 3.2 at page 4, and Section 3.4 at page 5.
> 3. **Dataset Characterization**: This change is raised by Reviewer 644A discussed in *To Comment 2*. We detailed dataset characterization with 5 statistics in Appendix A.4, page 29, with a footnote in Section 4 at page 5.
> 4. **More Failure Modes Examples and Highlighting**: This change is raised by Reviewer xPFt discussed in *To Comment 3*. We added 4 more examples and highlighted reasoning failures with red boxes in Appendix A.2 at pages 21-25, with footnotes in Section 5.1 at page 8 and 9.
> 5. **More Limitation Discussion**: This change is raised by Reviewer 644A discussed in *To Comment 3*. We revised it in the Limitations section at page 12.
> 6. **All figures in PDF format**: This change is raised by Reviewer xPFt discussed in *To Comment 1*. We have re-exported all figures in PDF format with no pixelation.